# Intelligent Knee Sleeves: A Real-time Multimodal Dataset for 3D Lower Body Motion Estimation Using Smart Textile

**Wenwen Zhang**[1][*] **Arvin Tashakori**[12]**, Zenan Jiang**[12]**,Amir Servati**[2]**, Harishkumar Narayana**[2]**,**
**Saeid Soltanian**[2]**, Rou Yi Yeap**[2]**, Meng Han Ma**[2]**, Lauren Toy**[2]**, Peyman Servati**[12][*]

[1]Department of Electrical and Computer Engineering, University of British Columbia
[2]Texavie Technologies Inc.
{wenwenzhang, arvin, jiang, peymans}@ece.ubc.ca
{aservati, harishkumar, ssoltanian, ryeap, meganma, ltoy}@texavie.com

## Abstract

The kinematics of human movements and locomotion are closely linked to the activation and contractions of muscles. To investigate this, we present a multimodal dataset with benchmarks collected using a novel pair of Intelligent Knee Sleeves (Texavie MarsWear Knee Sleeves) for human pose estimation. Our system utilizes synchronized datasets that comprise time-series data from the Knee Sleeves and the corresponding ground truth labels from visualized motion capture camera system. We employ these to generate 3D human models solely based on the wearable data of individuals performing different activities. We demonstrate the effectiveness of this camera-free system and machine learning algorithms in the assessment of various movements and exercises, including extension to unseen exercises and individuals. The results show an average error of 7.21 degrees across all eight lower body joints when compared to the ground truth, indicating the effectiveness and reliability of the Knee Sleeve system for the prediction of different lower body joints beyond knees. The results enable human pose estimation in a seamless manner without being limited by visual occlusion or the field of view of cameras. Our results show the potential of multimodal wearable sensing in a variety of applications from home fitness to sports, healthcare, and physical rehabilitation focusing on pose and movement estimation.

## 1 Introduction

Attributed to the widespread adoption of machine learning (ML) methods in various domains, the field of computer vision has witnessed remarkable progress in the area of pose estimation [1]. These achievements, in turn, facilitate the development of activity recognition [2–4], point-to-point healthcare applications [5–7], augmented reality (AR) [8], and human-computer interactions [9]. Images and videos are usually the main sources for ML models to extract human pose, with major challenges including multi-person pose estimation, occlusion, and limited field of view (FoV) of cameras [10]. Moreover, concerns for data privacy in camera-based methods also encourage non-vision-based frameworks [11, 12] for human pose estimation that can provide more private data gathering. Since human motion must involve muscle activation, stretching, and contraction, we propose a pair of Smart Knee Sleeves with embedded yarn stretch sensors and Inertial Measurement Units (IMUs) to detect muscle contractions and joint movements, reflecting human movements. Recent advances in flexible electronics have demonstrated the feasibility of advanced wearable sensor

---

[*]Corresponding authors

37th Conference on Neural Information Processing Systems (NeurIPS 2023) Track on Datasets and Benchmarks.

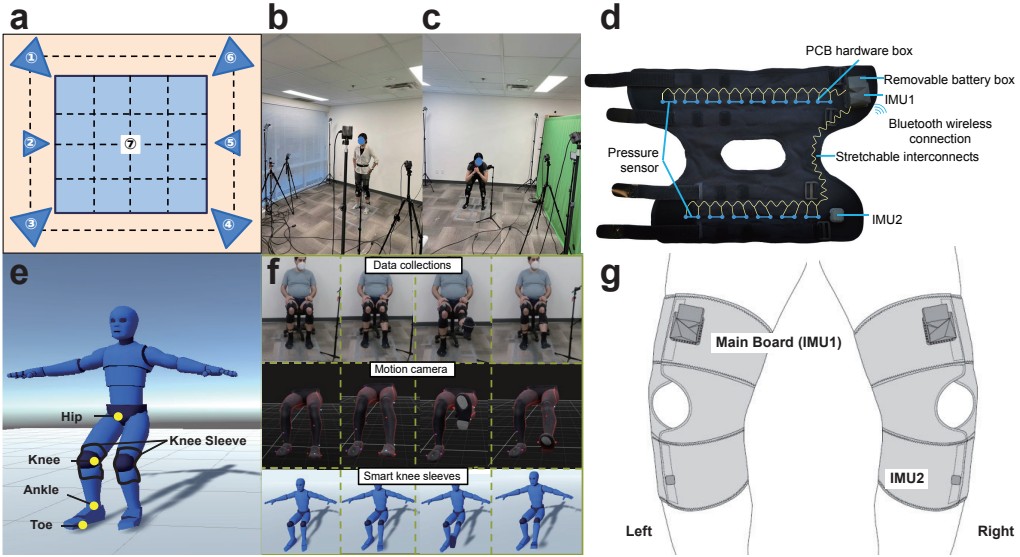

**Figure 1  Overall outline of the intelligent Texavie MarsWear knee Sleeves based 3D pose estimation process including the data collection, hardware setup, and qualitative results.** (a) Marker-based camera setup to capture major joint angles of the lower body during the exercises. The output time-series data recording joint movements will be used as supervised annotations in training steps. ①-⑥: MoCap cameras; ⑦: subject location for data acquisition. (b-c) Photographs of the experimental environment during data collection incorporating the wearable sensors. (d) An unfolded version of Texavie MarsWear Smart Knee Sleeve, displaying the location of the PCB hardware box, removable battery box, Bluetooth connection, stretchable interconnects, pressure sensors, and IMUs. (e) Major joints included in the training and testing process. (f) Visualization of the 3D human model for lower body pose estimation for both the MoCap camera system and smart Knee Sleeves. (g) Schematic of smart Knee Sleeves work by a user.

motion capture (MoCap) and pose estimation [13–16] with different form factors and performance parameters. Closing the gap between the current portable wearable devices' ability to estimate human posture and more accurate joint angle and movement estimation holds immense potential for facilitating healthcare applications, aiding individuals with joint-related illnesses (such as arthritis, rheumatism, or osteoporosis), as well as assisting in sports analysis [17, 18].

In this research, we introduce a comprehensive dataset with extensive ground truth labels from MoCap camera systems and a baseline model for pose estimation tasks based on an overview architecture as displayed in Figure 1. Here, Figure 1 (a) depicts the camera setup for our MoCap system, which provides the ground truth labels for our supervised learning. Figure 1 (b-c) show the data collection process displaying the relative position of the subject wearing the Knee Sleeve device and cameras. Smart Knee Sleeves (provided by Texavie) are crafted from stretchable and washable textile materials (Weft knitted double-jersey rib fabrics composed of polyester/spandex) as shown in Figure 1(d) and (g). The smart textile device is embedded with yarn-shape pressure sensors located around the hamstring and quad muscles as well as calf and shin muscles on the legs of the user and two IMUs above and below the knee joints. Wavy 3D stretchable interconnects connect all sensors and IMUs to a wireless readout and processing board with a rechargeable battery.

We monitor four major joints of each leg (hip, knee, ankle, and toe) on the left and right sides separately, using MoCap system. Our Smart Knee Sleeves provide 14 channels of pressure sensor data, indicating muscle contractions related to movement, and 9 channels of IMU data that capture the angle of the knee joint. The unfolded version of our knee sleeves is schematically displayed in Figure 1 (d), showing the location of the PCB, embedded pressure sensors, removable battery box, IMUs, and stretchable interconnects. Our smart knee sleeve and custom-made software developed by Texavie Technologies Inc, work together through a special wireless communication system, enabling us to record real-time reactions of muscles and joints during exercise and movements as displayed in

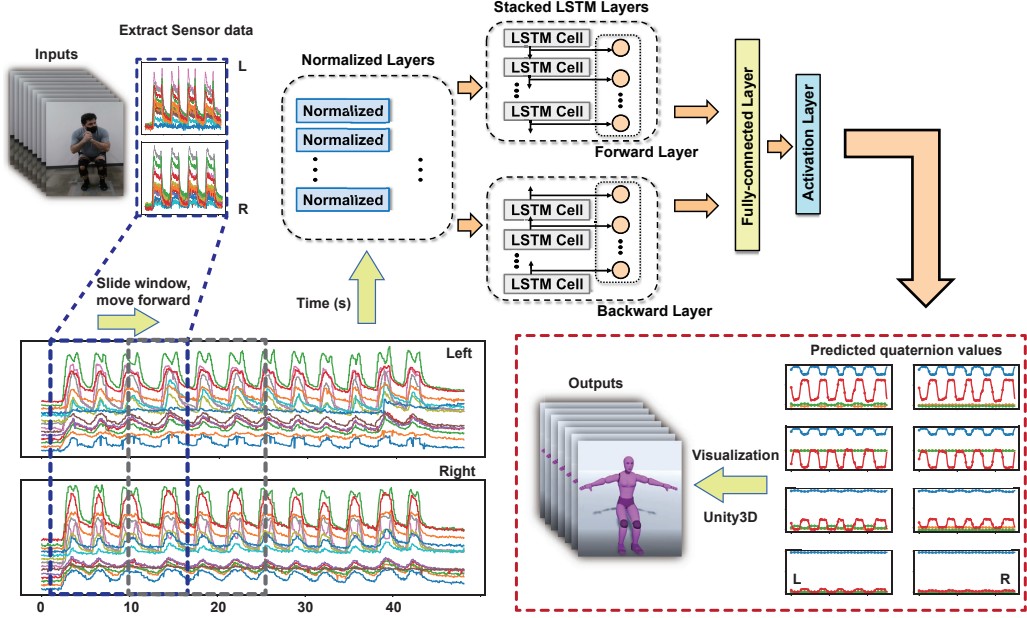

**Figure 2  Architecture of the 3D pose estimation machine learning (ML) model.** The baseline ML model architecture utilized in this work to estimate joint movements. The input is sensor signal readouts with a sliding window from our smart Knee Sleeves, and the output is the joint motion in quaternion. We visualize the output quaternions through a Unity3D human model.

Figure 1 (d). Developed iOS app and supporting software to enable easy collections of various daily exercise poses from the Knee Sleeves. With the assistance of MoCap guidance, we have developed a recursive neural network-based model that can estimate 3D human lower body joint angles by fusing data from IMUs and pressure sensors, as illustrated in Figure 2. The neural network utilizes normalized sliding-window sensor fusion data from our smart Knee Sleeves to generate real-time time-series quaternions that estimate the motion of all joints of the lower body. The 3D human model visualization is developed in Unity3D. The qualitative visualization results in Figure 3 exhibit comparative outcomes from RGB images during data collection, ground truth quaternions extracted from the MoCap system, and estimated 3D human model visualization results from Knee Sleeves.

Our ML model under the supervision of the commercialized MoCap system information, can accurately predict the 3D human pose with an average joint angle error of 7.21 degrees, compared to the ground truth data obtained from the MoCap system. Furthermore, we evaluated the model's ability under different scenarios to generalize to new individuals and poses. The proposed smart Knee Sleeves can overcome the challenges of occlusion and multiple-person detection faced by camera systems. Through the use of smart textile force/stress sensing fused with IMU data, the proposed solution opens up possibilities for human pose estimation that is unaffected by visual barriers and can be executed seamlessly and privately. To the best of our knowledge, this is the first work to propose the prediction of lower body 3D human joint angles solely from a pair of customized, stretchable, wireless smart knee sleeves. To sum up, the principal achievements of this article include:

- A comprehensive multimodal dataset with synchronized wearable recordings of embedded pressure sensors, IMU, and marker-based MoCap data on major joints of the lower body.

- A baseline model on time-series data for 3D predictions of major joints on the lower body with an average of 7.21 degrees, going beyond the knee joints and to other joints using the smart Knee Sleeve. The public access to our synchronized dataset and baseline model is at https://feel.ece.ubc.ca/smartkneesleeve/.

- Extension and generalization of our prediction model to unseen exercises and individuals.

The following is the organization of the paper: Initially, we provided a summary of current 2D and 3D human pose estimation techniques, followed by an investigation of proposed benchmarks for

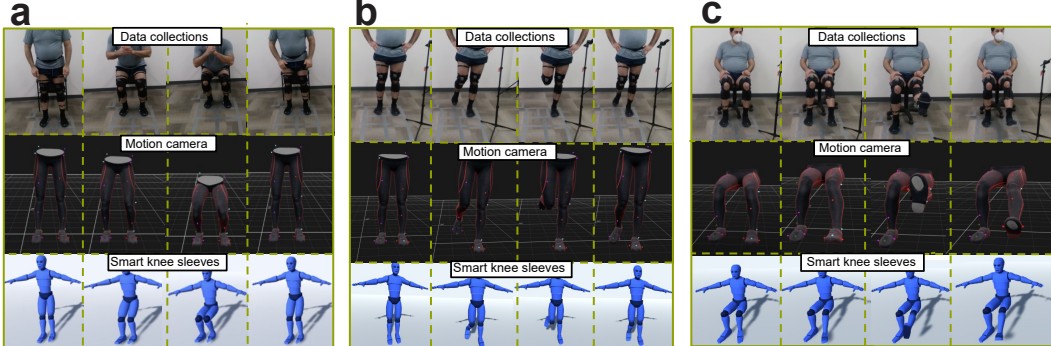

**Figure 3   Qualitative Results of Smart Knee Sleeves Across Time Steps.** The depicted poses, from left to right, are squatting (a), hamstring curling (b), and leg raising (c). For each sequence, from top to bottom, we showcase our data collection setup, ground truth annotations captured by the MoCap, and the qualitative outcomes derived from our knee sleeve readouts displayed using a human model in Unity3D.

kinesthetic sensing in textile-based wearable sensors in section 2. Then, we presented the specifics of our smart wearable sensor dataset in section 3, including how we acquired and pre-processed the data. Afterward, we described the implementation details, baseline models, and performance for our dataset in section 4. Following that, we discussed the limitations of our baseline model and analyzed their causes in section 5. Lastly, we wrapped up the paper in section 6 and included supplementary materials for additional information.

## 2   Related Work

### 2.1   Human Pose Estimation

The presence of emergent human pose datasets and the introduction of deep neural network models have led to significant advancements in human pose estimation from images or videos in recent years. MoCap system are used as ground truth for these studies. As shown in Figure 1 (a-c), we used reliable MoCap systems (Optitrack) to provide supersized annotations in our training process. We used six cameras around the subject (marked by 7) to fully capture the motion in 3D planes as shown in Figure 1(a). Calibration is required every time before data collection since the relative location between the subjects, markers, and cameras will have a great influence on the final outputs from camera-based algorithms. Key-points estimation on joints to predict human pose [19–23] has been a popular method in the human pose inference area. Multi-view [24], special data augmentation [25] or multi-modal data [26, 27] are usually required to assist in the prediction of 3D key points with vision and camera-based methods. Subsequently, the demand to extract more detailed information about the human body's posture and movements has driven the interest in 3D pose estimation utilizing 3D human models [28–30]. Pose estimation with 3D human models are capable of providing more details on the orientation of the body joints, skeletal structure, etc, and thus is more resource-intensive in computer vision tasks.

To capture more detailed information about joint angles and movements while requiring lower computational resources, wearable sensors have emerged as a promising alternative to camera-based methods for 3D pose estimation. Camera-based methods largely rely on visual cues to infer the position and orientation of body joints, and face many challenges including fixed equipment location [31, 7], lighting conditions [8], environment, background noise [32, 21], occlusion [33], and multi-person problems [34, 35]. Wearable sensors, on the other hand, avoid these issues as they don't require a clear and unobstructed view of the body. Meanwhile, flexible electronics can provide real-time measurements of the dynamic movement status of the human body segments and are a reliable source of kinesthetic information under a wide range of conditions, including outdoors, low-light, noisy, and cluttered environments.

IMUs-based kinesthetic sensing [36–40] excel in wearable pose estimation primarily due to their self-contained operation, eliminating the need for external references or beacons. Their compact

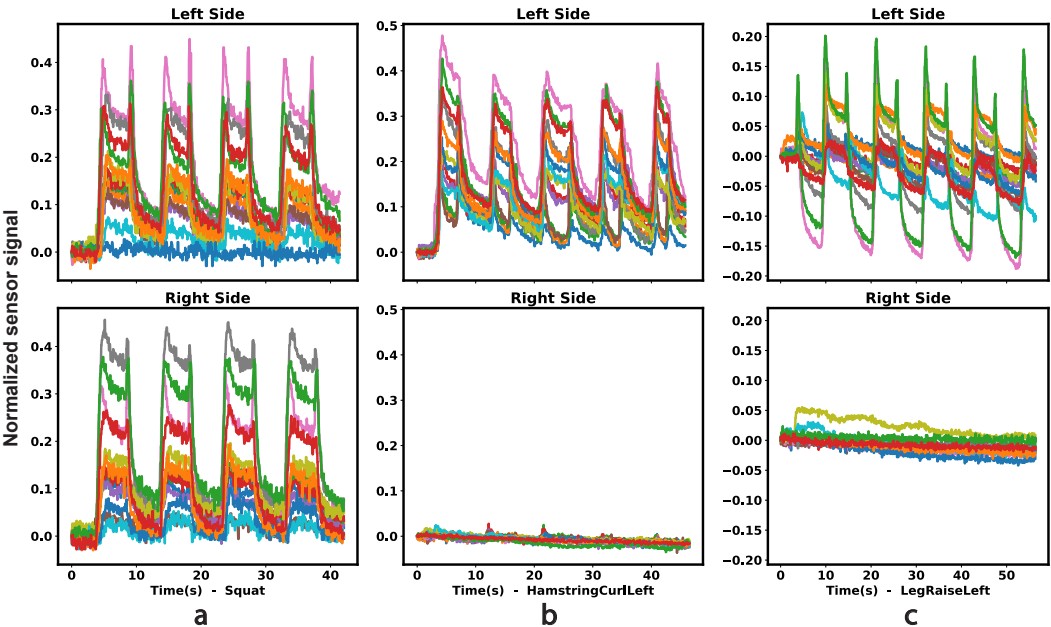

**Figure 4  Normalized sensor signal during different exercises.** The exercises from left to right are (a) squat, (b) hamstring curl, and (c) leg raise, respectively. The electric signals generated by the pressure yarn sensors correspond to the degree of stretching and muscle contractions. In the absence of movement in a resting leg, flat lines for the right leg for the hamstring curl (bottom sub-panel b) and leg raise (bottom sub-panel c) are shown for better comparison, where only the left leg is intentionally moving. This describes the kinematic process yielding the sensor output depicted within the illustrated diagram.

design ensures user comfort, while their capacity to integrate with other sensors, like magnetometers, boosts accuracy and mitigates drift. [40] employ IMU-based equipment positioned on the head and hands to predict comprehensive full-body poses in Mixed Reality, which overcomes the constraints of existing systems that provide only partial virtual representations. [37, 38] aim to efficiently predict precise human poses with a mere six strategically placed IMUs (XSens) on the body, addressing the complications associated with traditional dense configurations and meeting the rising needs of interactive technologies. However, using solely IMUs for pose estimation faces some essential challenges, notably the drift errors that accumulate during position calculation by velocity integration or orientation determination by angular velocity integration. Supplementary technologies such as Kalman filtering, sensor fusion with other systems, or periodic recalibration are imperative to achieve optimal accuracy.

Our research goes beyond traditional vision-based and standalone IMU methods in adeptly detecting subtle, real-time changes in joint angles and movements. The Smart Knee Sleeves integrate both IMUs and pressure sensors to reduce drift errors effectively. These sleeves are convenient and designed for everyday wear, eliminating the need for any additional equipment to monitor daily activities and exercise routines. We deliver real-time 3D human models with details on 8 major joints of the lower body, which are immensely valuable for sports therapists to provide feedback on athletes' technique, rehabilitation [41] for tracking the progress of patients undergoing physical therapy [42] and enhancing human-computer interaction (HCI) [43, 44] and virtual reality (VR) experiences [45]. Those applications extend beyond the realm of mere 3D human pose estimation, benefiting various facets of society.

## 2.2  Kinesthetic Sensing through Smart Textile Fabric

Advancements in stretchable smart textiles [46, 47] have enabled the development of wearable devices that are well-suited for dynamic tracking, monitoring, and modeling of human movements in a variety of contexts. With the ability to detect kinesthetic feedback during body movement via detecting

force-induced deformations in muscle activation [48], stretchable smart textile modalities hold great possibilities for predicting 3D human pose with great accuracy. Designs for detecting human activities through textiles have been investigated in several major ways: producing electrical signals through human-environment contacting [14, 49–51], pressure change [5], and material deformation [6]. With those characteristics, textile-based sensors have been fabricated as wearable apparel/garments to wear on diversified parts of the body such as face [52], arms [53, 54], and hands [51, 55], to capture the dynamic status of the designated area. Luo et al. [13] predicted the key points of joint angle from tactile carpet, which partially solve multi-person problems. However, their data include large-scale no interaction data, where no pressure is produced with no subject standing. Zhang et al. [51] proposed e-textile gloves to sense contact with objects and movement, which also include a large portion of background data and are unable to provide joint details of the finger. Xu et al. [53] adopt responsive pressure sensors to detect arm movement, but they focus on classification tasks only.

Distinguished from previous work focusing on wearable sensors, which are mostly classification tasks or unable to provide direct information on joint angles and motions, we aim to predict major joint movement in the lower body with subtle details of orientation and bending in three directions as illustrated in Figure 1 (e,f). The pressure sensor and IMUs are designed around the thigh and calf regions to capture the kinesthetic feedback from different orientations. We provide complete pipelines from ML-based joint prediction to human 3D model reconstruction with the assistance of Unity3D as depicted in Figure 2.

## 3    Smart Wearable E-textile Sensor Dataset

**Data Acquisition**    Our stretchable knee sleeves include 14 channels of sensor arrays and 9 channels of IMU data from two Bosch Sensortec BNO055 IMUs. A customized readout circuit board is designed and fabricated by Texavie Technologies Inc., to arrange and fuse multiple channels of data from both pressure sensors and IMUs, and to capture subtle changes around major joints during exercise. Paired with specialized mobile software constantly communicating with the hardware through Bluetooth low energy protocol, we are able to acquire data free of wires and realize the real flexibility and wearable to track human movements. Our smart Knee Sleeves are personalized, robust, and highly reliable for data collection under various physical conditions and exercises. Under the paired Bluetooth connectivity, we acquire over 300 sensing readouts at a 20 Hz sampling rate for the left and right knees. As shown in Figure 4, our smart Knee Sleeves exhibit high responsiveness to changes in muscle contraction and relaxation during exercise poses. The pressure sensors remain stable in the absence of external stress or deformation. During exercises such as squats, hamstring curls, and leg raises, the pressure sensors on both the left and right knee generate electric signals that correspond to the level of stress sensed at designated locations. In the case of the squatting pose, the left and right knee signals are similar due to the comparable muscle reactions on both sides of the body, carrying information about the symmetry of movement and muscle forces. For the hamstring curl and leg raise poses, we observe more significant pressure sensor responses on the left side than the right side as it serves as the primary exercise leg.

Using the multi-modal data from multiple channels, we are capable of estimating angles for major joints of the lower body during subject's movements. We have acquired over 140,000 synchronized frames of data from our stretchable wearable smart textile modality and MoCap system from 12 continuous days from different subjects with various sizes of Knee Sleeves. The details of subject numbers, task numbers, and other details are summarized in Table B6. For ethical considerations, please refer to Appendix C.

**Data Pre-processing and Augmentation**    We extract ground truth data from the MoCap system, where markers are required to calculate joint angles. We calculated the relative angles of the joints from the MoCap system and used these angles as supervised labels in the training task. The output label contains 8 joints' time-series quaternions for the left and right legs, respectively, as illustrated in the label-generation process Figure A4 of Appendix A. The details including the content, structure, and dimension of our dataset are summarized in Appendix B. This is an example of generating time-series labels from a squatting exercise. It is important to recognize that occlusion problems can impact MoCap systems, leading to inaccuracies (refer to Figure A1 for details) in ground truth labels. As a result, this can generate errors during subsequent training procedures. But this error is not caused by our model or wearable devices and can be alleviated by removing unreasonable

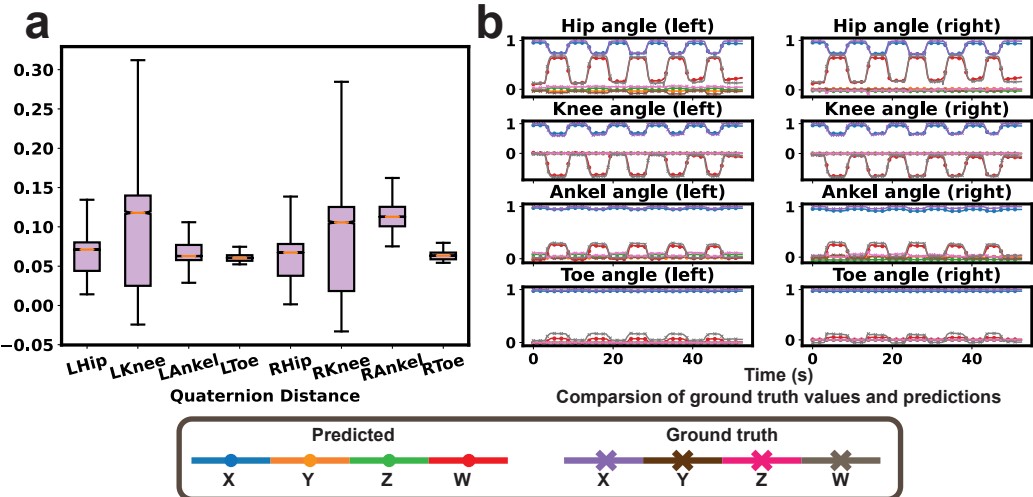

**Figure 5  Quaternion distance and estimation results comparison**. (a). The model's overall performance evaluated on the entire dataset, encompassing all exercises and individuals. (b). The quaternion output from the models. The knee angle prediction showed the highest level of accuracy across all joints. The toe angle was found to be mostly stable with minimal movement during the squat exercise.

ground truth annotation data. But as the baseline model, we incorporated the entirety of the MoCap system's collected data to ensure data integrity. This also verifies that wearable sensor integrative smart sleeve benchmarks are more accurate and reliable than computer vision methods under certain circumstances where occlusions occur.

Although Bluetooth and wireless communication have contributed to the development of flexible and mobile devices for use in daily activities and exercises, the latency of Bluetooth [56] may cause uneven time intervals. Similarly, we observed uneven time intervals for our smart knee sleeves as well, whereas the MoCap system consistently provides an evenly increased time axis. To align the data from the MoCap system with the output from the Knee Sleeves, we employ the Fourier method to resample the Knee Sleeve readouts.

**Table 1  RMSE in degrees for smart Knee Sleeves performance evaluation on various scenarios.** The first row is RMSE for all seen tasks, while the second to fourth rows are RMSE for unseen squats, hamstring curls, and leg raise exercises, respectively. Refer to Table A3 for more details.

| Scene | Pose | LHip | LKnee | LAnkel | LToe | RHip | RKnee | RAnkel | RToe |
|-------|------|------|-------|--------|------|------|-------|--------|------|
| All_seen | Avg | 9.03 | 11.80 | 6.23 | 3.81 | 9.31 | 7.69 | 7.04 | 2.77 |
| Unseen Tasks | BendSquat | 17.50 | 14.20 | 12.30 | 4.25 | 17.90 | 15.10 | 12.10 | 5.12 |
| | Hamstring Curl | 12.70 | 18.00 | 6.13 | 2.71 | 12.40 | 16.90 | 6.49 | 4.13 |
| | Leg Raise | 10.20 | 19.80 | 9.05 | 2.56 | 9.55 | 16.20 | 9.29 | 5.50 |

## 4  Implementation Detail and Experimental Results

**Implementation**   We implement the baseline neural network using 2 layers of long short-term memory (LSTM) with Pytorch [57], as shown in Figure 2. We use data from the pressure sensors and IMUs as input and that from the MoCap system as ground truth labels to train the LSTM model. The output from our ML model is the quaternions of the eight major joints of the lower body. The sequence length used to create the sliding window is 250 sample index to capture the change of pressure sensors and IMUs during movement. We choose the tanh function as activation to match the output range of quaternions from -1 to 1. All sensor readings and quaternions should be normalized

to range (-1,1) before training to bring features to a similar scale. We ran the experiment on NVIDIA GeForce RTX 2060 and got results within 5 hours.

**Experiments**    We trained our ML model on 109,000 pairs of MoCap and wearable sensor output frames and validated and tested on 13,000 frames of data. We evaluated the results with quaternion distance ($D_q$) to compare the estimated 3D joints' angles with corresponding values from MoCap ground truth data, as shown in Figure 5 (a). The calculation of $D_q$ (see Equation 1 for details) is performed under the scale of normalized quaternions [58]. Figure 5 (b) illustrates an example output of quaternions for the squat exercise separately derived for the left and right legs. To enhance comprehension, the evaluation results expressed in Euler angles is incorporated into the Figure A6. The motion recorded by the pressure sensors aligns well with the changes of quaternions, as displayed in Figure 4. Table 1 summarizes the root-mean-square error (RMSE) of each joint in degrees converted from quaternion distance results. We report average errors of 9.16, 9.75, and 6.64 degrees for the knee, hip, and ankle angles, respectively. The toe has a relatively low margin of error because it is not a primary joint used in squats and is not significantly involved in activities during exercise.

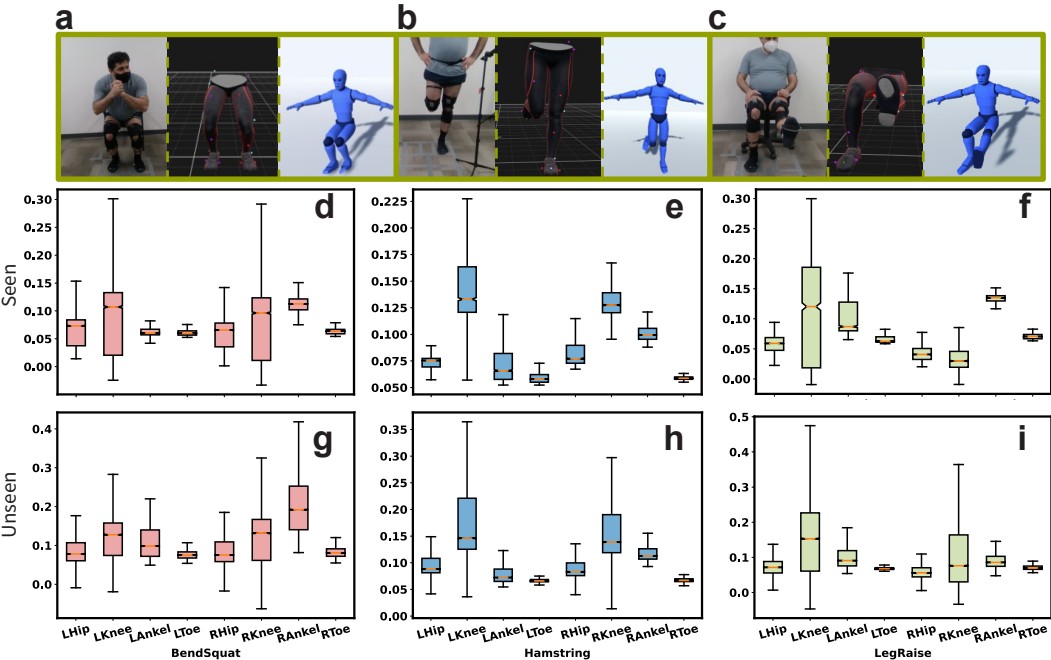

**Figure 6    Device's generation evaluation (unit: normalized quaternions). Top (a-c)**: Qualitative performance extracted from unseen tasks for squat, hamstring curl, and leg raise exercises. **Middle (d-f)**: Error of joints for predictions of seen tasks. The model sees all the data from participants and tasks. **Bottom (g-i)**: The occurrence of prediction joint quaternion error in new tasks and individuals. The training was conducted on a partitioned dataset that excluded tested actions to exam the generalization of our model to unseen tasks.

Our assessment involves measuring the device's ability to estimate joint angles for activities that have not been previously observed. As displayed in Figure 6 (g-i), the model performs well on different unseen tasks. Our dataset is roughly categorized into three types of exercises: squat, hamstring curl, and leg raise. Despite the various forms of squatting available (stepwise squat, tired squat, etc. See Table B7 for details), we view them as the same movement when it comes to training. In our trials of unseen exercises, we exclusively evaluated the bent squat Figure 6 (d, g) because other variations of squatting produced very similar results. To estimate the bent squat, we trained solely on exercises involving leg raises and hamstring curls, excluding all other types of squats from the training process. In theory, regardless of the type of exercise performed, the pressure sensor and IMUs should exhibit comparable patterns as long as there is similar muscle contraction and extension since the human pose is essentially linked to muscle activation.

Our smart knee sleeve generalizes to unseen poses with slightly increased errors as for the case of hamstring curls Figure 6 (e, h). The reasonable degradation of performance in unseen tasks can arise from the mildly distinctive patterns in leg raise. Leg raise in Figure 6 (f, i) is the only pose in our dataset that starts from a sitting position. Since we are measuring the pressure sensor and IMUs with relative values to avoid the sensor and marker displacement influence, the start point for both strain sensors and IMU data is zero. This is reasonable for the poses that start with the standing position. However, for the sitting position, the supervised labels provided by the MoCap are actually 90 degrees, and the pressure sensors will also have initial values with stress applied. To eliminate the effect, we rotate all the quaternions of leg raise from IMUs 90 degrees before training. The pressure sensor data will also have a relatively influenced pattern due to the initial sitting position. The inconsistency between IMUs, sensors, and ground truth data induces confusion and errors in the model inference process. If we let the model see only 10% of the leg raise data in the training process, the performance will be improved with less error (see Figure A5 of Appendix A).

Our wearable Knee Sleeves are customized to fit each individual perfectly. Except for poses that have not yet been encountered, it is possible that the wearable electronics, markers, and MoCap system calibration positions may shift when tested on different dates and individuals. We have conducted tests under these conditions and have depicted the results in Figure 7. No discernible decrease is observed in the outcome and accuracy of the model.

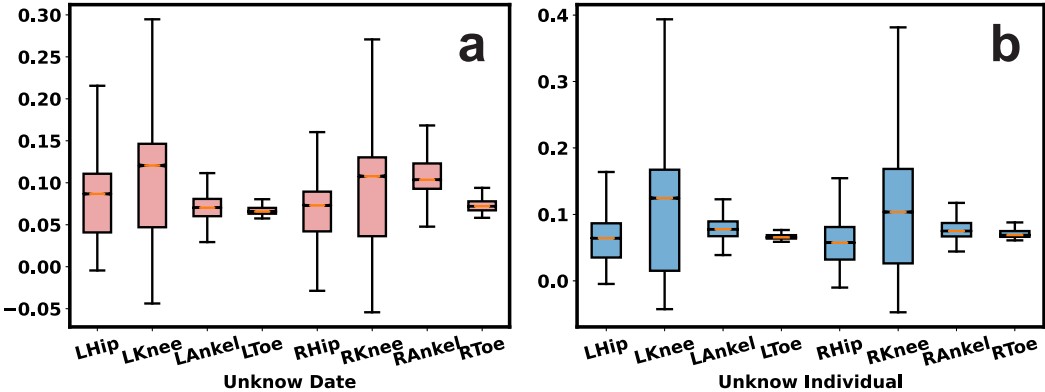

**Figure 7  Quaternion distance for unknown individual exercises (a) and unseen dates (b).** The model's performance remains consistent when trained with unknown individuals and dates, with only a minimal rise in the quaternion distance error, indicating its strong generalization capabilities.

## 5  Discussions and Limitations

Our attempts to obtain precise angle measurements from the lower body's anatomical pose have encountered multiple challenges that can compromise measurement accuracy during exercises. These challenges include soft tissue movement and sensor displacement during prolonged exercises, which can result in potential errors. Moreover, the accuracy of our model inference is compromised when testing the leg raise pose, which is the only pose starting from a seated position. To overcome these challenges and enhance our system, we will include additional scenarios that involve transitioning from sitting to standing or lying down to examine the impact of the starting position on our smart Knee Sleeves measurements. We plan to modify the starting position from relative to absolute values or add a calibration period to ensure accurate measurements across all poses by aligning sensor values at 0. Furthermore, when measuring errors from various joints, we noticed that the toe angles consistently demonstrate low error rates. This is likely due to minimal movement in the toe angle during these poses. To better evaluate and compare the model's performance on joints, it would be preferable to use percentage error measuring systems or add poses that include obvious toe movement.

In addition, we discovered that MoCap systems can encounter occlusion problems, which can affect the accuracy of ground truth labels in our dataset. As a result, we plan to thoroughly distill the ground truth data and ensure that as supervised information, MoCap feedback is accurately interpreted to achieve improved accuracy in the future. What's more, we have mentioned using Fourier resampling

to make the recordings from wearable knee sleeves more smooth in section 3, but Fourier transform can't thoroughly solve the uneven time interval problem, which may cause drifted predictions in the test as in Figure A3. Future methods will be recommended to include more specific and complex algorithms focusing on lost time points to address Bluetooth issues.

## 6    Conclusions

We provide a comprehensive dataset and baseline model for 3D human pose estimation with a pair of durable, stretchable, wearable sensors. We demonstrate our ML model pipeline's effectiveness across various scenarios including generalization to unseen tasks and individuals. We collected a synchronized dataset that comprised time-series data from our smart Knee Sleeves and corresponding ground truth labels from MoCap system. By utilizing these perception outcomes as guidance, our system was able to generate 3D human models solely based on the wearable sensor-integrative apparel readings of individuals performing diverse activities. We achieved an average RMSE of 7.21 degrees across eight joints in the lower body compared to commercially available MoCap tools. Our work offers a novel sensing modality that complements traditional vision systems and enables human pose estimation without being impacted by visual obstructions in a seamless and confidential manner. This innovation has potential applications from home fitness to sports analysis, personalized healthcare, and physical rehabilitation focusing on pose and movement estimation.

## Acknowledgments and Disclosure of Funding

The smart Knee Sleeves and related app and software for data readout are provided by Texavie Technologies Inc. Texavie collects all wearable sensor data we analyzed in this paper. We express our gratitude to the volunteers who participated in the data collection experiment, as well as to the anonymous reviewers for their valuable comments and discussions. This work received partial support from the University of British Columbia. The opinions, findings, conclusions, and recommendations presented in this paper belong to the authors and do not necessarily represent the views of the funding agencies or the government.

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
