# OpenReview forum: "Intelligent Knee Sleeves: A Real-time Multimodal Dataset for 3D Lower Body Motion Estimation Using Smart Textile"
_NeurIPS.cc/2023/Track/Datasets_and_Benchmarks — NeurIPS 2023 Datasets and Benchmarks Poster_

### Official Review · Reviewer_9KjM · 2023-07-21
**Review of Intelligent Knee Sleeves**

**Rating:** 6
**Confidence:** 4
**Correctness:** Yes
**Clarity:** The paper is easy to follow, but some…

**Strengths:**

This paper presents a multimodal dataset for lower body motion estimation with synchronized wearable recordings of embedded pressure sensors, IMU, and marker-based MoCap data, which may benefit the research in human pose estimation.

**Additional Feedback:**

See above

**Documentation:**

Yes

**Limitations:**

Yes, the authors addressed the limitations such as challenges caused by soft tissue movement and sensor displacement, occlusion problems, and drifted predictions and discussed the possibilities for future work.

**Opportunities For Improvement:**

1.Since it is a multimodal task, it is important to show the performance of each device and how the multiple sensors can help each other, but such a study is missing.

2.Only rotation error is listed as a evaluation metric. However, in human pose estimation task, mean per joint position error (MPJPE) is used. It is also necessary to measure the smoothness of predicted results.

3.There are already some related works for lowe-body/full-body pose estimation from sparse sensors, like [1,2]. It would be better to use these existing work as baseline models, or at least discuss about the difference between them and the presented work.

[1] Avatarposer: Articulated full-body pose tracking from sparse motion sensing, ECCV 2022

[2] Deep inertial poser: Learning to reconstruct human pose from sparse inertial measurements in real time, TOG 2018

**Relation To Prior Work:**

Since there are already work based on full-body/lower-body pose estimation from sparse sensors, it is important to discuss their difference, and show how the added sensors can help the estimation.

**Summary And Contributions:**

The paper introduces a multimodal dataset obtained using Intelligent Knee Sleeves for human pose estimation. The authors propose a baseline model utilizing time-series data to predict 3D positions of major lower body joints, achieving an average error of 7.21 degrees. Additionally, the generalization ability of the proposed model is evaluated through tests on unseen exercises and individuals.

---

> ### Author Response · Authors · 2023-08-15
> **Answer to Reviewer 9KjM**
>
> - **Table A3: Mean Squared Derivative results for quaternion predictions on all orientations.**
>
> | Orientation | x | y | z | w |
> |:---:|:---:|:---:|:---:|:---:|
> | MSD | 6.92e-05 | 9.82e-09 | 7.08e-09 | 5.46e-04 |
>
> Dear reviewer, thanks for your valuable and helpful comments on our paper.
> - We utilize both IMUs and pressure sensors to enhance data measurement accuracy and ensure more reliable pose estimation tasks. For a detailed comparison of how multimodal data improves accuracy, please refer to Table A2 in our collective response to all reviewers. Both IMUs and pressure sensors greatly enhance the final model performance. The use of an integrated device that offers fused multi-modal data substantially improves the accuracy of the estimation.
> Concerns on the joint location error: Our research is specifically focused on joint rotation prediction, which is a critical aspect of monitoring joint health, especially for patients with joint-related conditions. [6] focus on the whole body pose estimation through equipment on the upper body (headset and two handheld controllers), and evaluate their model performance through MPJPE. They emphasize more on comprehensive coordination all over the body to achieve a more vivid performance in VR. We noticed a lot of recent work that either provides local error evaluation [3] or joint angle rotation error evaluation [4,7], depending on their specific methods. In our work, we focus on precise joint angle rotation estimation and related specific healthcare applications such as musculoskeletal patients, therefore, we emphasize the precision of rotation angle error over location error. From the author’s knowledge, our work is the first to focus on precise joint rotation estimation with ML-based Smart Knee Sleeves. This has great meaning for certain healthy applications focusing on specific diseases that suffer from incorrect joint rotation. We summarized related work in Table A1 for a comparison that highlights our contributions.
> - We agree that smoothness is an important index in the time-series prediction scenario. We provide the Mean Squared Derivative (MSD) of our predictions in Table A3. Our prediction data are quaternions scaled from 0-1, and the MSD for each orientation is at the scale of  10^-4, which proves our prediction results don’t have abrupt changes and are smoothly transitioned. We attached the smoothness analysis in lines 88-91, P5 of the Appendix.
> - We appreciate the valuable papers you provided. All of them are exceptional and contribute significantly to our understanding of the field. For Jiang et al. 2022 [6] and Huang et al. 2018 [8], we have incorporated their findings into our research, particularly in the realm of pose estimation through wearable sensors, and cited them in the related work section of our paper (P5 line 117-139), enriching the diversity of our related work. Meanwhile, we are exploring additional applications that could potentially benefit from our device. However, it's important to note that the aforementioned papers primarily focus on the use of individual IMUs attached to separate parts of the body for pose estimation and human movement analysis. In contrast, our work emphasizes the use of integrated wearable and stretchable devices embedded with pressure sensors. This approach allows us to gather more accurate data, which enhances the precision of our analyses on focused joints with fewer IMUs and more integrated devices. After training our model, there's no longer a need for a Mocap system, making the setup simpler and more user-friendly. Our calibration protocol can be streamlined by focusing on the collection of diverse data, a less complex process compared to the intricate IMU systems described in [8]. The incorporation of more IMUs necessitates more complex calibration protocols. Our pressure sensors, located around the thigh and shank muscles, can directly detect local deformations, including stretches and pressures in the fabric caused by joint movements. The use of state-of-the-art materials allows for a direct reflection of lower body movement, eliminating the need for complex sensor fusion algorithms. The work in [6] primarily focuses on the interaction between the headset and two handheld controllers. Our aim, however, is to support healthcare applications, assist individuals with joint-related illnesses such as arthritis, rheumatism, or osteoporosis, and aid in sports analysis. This necessitates a more detailed focus on knee joints, as opposed to purely 3D human pose estimation tasks. While we acknowledge the significance of 8 in the field of gaming and Virtual Reality (VR), it is challenging to use their results as our baseline given our primary focus on digital health and wellness experiences for musculoskeletal patients at home or in clinical settings.

---

> > ### Comment · Reviewer_9KjM · 2023-08-29
> > **Respond to Answer to Reviewer 9KjM**
> >
> > Dear authors,
> >
> > thanks for your detailed response. I appreciate your efforts and some of my concerns were solved, so I would like to raise my score. However, there are still two points that need to be considered:
> >
> > 1. This paper is not written in a way that focuses on healthcare application, but on general human pose estimation, so it is always necessary to have the position error as a metric.
> >
> > 2. Mean squared derivative seems to be a wrong metric to measure smoothness. Instead, jerk (derivative of acceleration) is usually used to measure jitter.

---

> > > ### Author Response · Authors · 2023-08-31
> > > **Reply to Reviewer 9KjM**
> > >
> > > Dear Reviewer,
> > >
> > > We genuinely appreciate your openness to the clarifications and additional data provided during the rebuttal process. We extend our heartfelt gratitude for your thoughtful reconsideration.
> > >
> > > Regarding the Focus on Position Error:
> > > Our research is engineered to provide a comprehensive solution for 3D lower body motion estimation, specifically optimized for healthcare applications. Despite its specialized focus, our approach is sufficiently versatile to be applicable in a diverse array of contexts. To gauge the effectiveness of our system, we employ rotation error as our primary evaluation metric, which we consider to be a meaningful and adequate measure within our targeted domain. Our model's final output consists of quaternions that accurately represent the orientation of targeted joints. This choice is intrinsically linked to the nature of our dataset, which comprises data collected from pressure sensors affixed to the thigh and shank regions to record real-time reactions of muscles and joints during exercise and movements.
> > >
> > > Our research is particularly concentrated on the rotational aspects of lower body motion, aiming to benefit healthcare applications. It is designed to assist individuals suffering from joint-related conditions such as arthritis, rheumatism, and osteoporosis, as well as to contribute to sports performance analysis where rotation errors better describe the model performance. Meanwhile, we recognize that position error is a more universally applicable metric in the broader realm of pose estimation, and we plan to explore its significance in our future works on this topic. Additionally, we encourage researchers in other domains to investigate a wider range of output representations and error evaluation metrics based on our benchmark. We are immensely grateful for your insightful feedback on our work and look forward to further collaboration and discussion.
> > >
> > > Regarding the Use of Mean Squared Derivative: In our research, we have chosen the MSD of quaternions as our principal evaluation metric, primarily because it aligns closely with our core objective of angle estimation. This metric offers an effective means to quantify the smoothness of angular movements, an attribute that holds significant relevance to our specialized application. While many studies focusing on acceleration and velocity commonly employ jerk to measure jitter, this metric may not be directly relevant to our specific area of investigation. In the current phase of our research, we didn’t feed acceleration data into our models. Instead, we have chosen to concatenate quaternions with pressure sensor data, owing to their correlated trends. Our 9-axis IMUs collect data from accelerometers, gyroscopes, and magnetometers—abbreviated as 'acc', 'gyro', and 'mag' for convenience in later discussions. It is worth mentioning that we concatenate quaternions, rather than raw sensor data, with the pressure sensor data. These quaternions are derived by fusing 'acc', 'gyro', and 'mag' data. Before initiating the training process, we normalize the pressure sensor data to fit within a range of -1 to 1, thus matching the range of the quaternions. Our preliminary analyses on the MSD of quaternions indicate that the smoothness of quaternion transitions supports the notion that our predictive outcomes are devoid of abrupt changes. Therefore, in our specific context, the derivative of acceleration appears to be superfluous and also irrelevant considering the contributions of our pressure sensors in estimating the quaternions of major joints of the lower body. In our multimodal dataset, we offer raw data from the acc, gyro, and mag sensors to support the needs of training diverse neural network architectures in the future. If you prefer to examine the derivative instead of the MSD of our predicted quaternions, we have included these details in Figure A6, located on page 7 of the SI. This figure therein further corroborates that our predictions are characterized by smooth transitions, devoid of abrupt changes.
> > >
> > > We hope that these explanations address your concerns. We appreciate your valuable comments on our paper.

---

### Official Review · Reviewer_Hbu8 · 2023-07-26
**Review on Submission685**

**Rating:** 6
**Confidence:** 4
**Correctness:** The claims made are believed to be co…
**Clarity:** The paper is generally well-written.

**Strengths:**

The idea of adopting smart knee sleeves for MoCap shed a new light on mitigating the occlusion issue of vision-based methods.

The trained ML model is shown effective with potential in generalization to unseen scenarios.



**Additional Feedback:**

N/A

**Documentation:**

The required documentation is provided.

**Limitations:**

It seems that only joint orientations are captured, while the locations are seemingly ignored, which could limit the generality of the proposed dataset and methods.

**Opportunities For Improvement:**

It would be helpful to investigate the contribution of different sensory channels to the errors. I'm curious on the contribution of the pressure data especially.


**Relation To Prior Work:**

Prior works have been well discussed.

**Summary And Contributions:**

Based on a novel set of intelligent knee sleeves, the authors proposed a multi-modal dataset to benchmark camera-free motion capture with the knee sleeves.
To collect the dataset, a MoCap system is built with cameras and wearable intelligent knee sleeves.
Four major joints of each leg are captured, with synchronized sensory data from knee sleeves.
A baseline ML model is trained to reconstruct lower-body motion given knee sleeve sensory data, which provides promising performance, showing the potential of knee-sleeve based MoCap.

---

> ### Author Response · Authors · 2023-08-15
> **Answer to Reviewer Hbu8**
>
> Thank you for your positive feedback and constructive suggestions. We appreciate your recognition of the novelty and potential of our work.
> - We agree that investigating the contribution of different sensory channels to the errors would be insightful, especially the contribution of the pressure data. We have included Table A2 in our collective response to all reviewers to elucidate the contribution of each modality to the final accuracy. As can be observed from Table A2, both IMUs and pressure sensors significantly enhance the final model performance. The use of an integrated device that offers fused multi-modal data substantially improves the accuracy of the estimation. We attached Table A2 with explains on lines 41-56 on page 4 of the Appendix, to elucidate the contribution of each modality within our multimodal dataset, emphasizing its impact on the overall accuracy of our system.
> - Thank you for your comment. Our research is specifically focused on joint rotation prediction, which is a critical aspect of monitoring joint health, especially for patients with joint-related conditions.  We noticed a lot of recent work that either provides local error evaluation [3] or joint angle rotation error evaluation [4,7], depending on their specific methods. In our work, we focus on precise joint angle rotation estimation and related specific healthcare applications such as musculoskeletal patients, therefore, we emphasize the precision of rotation angle error over location error. From the author’s knowledge, our work is the first to focus on precise joint rotation estimation with ML-based wearable sensors. This has great meaning for certain healthy applications focusing on specific diseases that suffer from incorrect joint rotation. We summarized related work in Table A1 for a comparison that highlights our contributions.
>
> Thank you for your feedback. We hope our responses have adequately addressed your concerns. We welcome any additional comments and suggestions you may have.

---

### Official Review · Reviewer_4Wwh · 2023-07-27
**Intelligent Knee Sleeves: A Real-time Multimodal Dataset for 3D Lower Body Motion Estimation Using Smart Textile**

**Rating:** 7
**Confidence:** 2
**Correctness:** The claims appear to be valid.
**Clarity:** Yes.

**Strengths:**

The problem of estimating body pose beyond the explicitly tracked body parts is a significant one. This data should be helpful to those working to estimate the lower-body configuration.

**Additional Feedback:**

NA

**Documentation:**

The documentation appears to be adequate.

**Ethics:**

No issues.

**Limitations:**

Limitations are discussed adequately.

**Opportunities For Improvement:**

NA

**Relation To Prior Work:**

Yes.

**Summary And Contributions:**

This paper presents a multimodal dataset for human lower-body pose estimation.

---

> ### Author Response · Authors · 2023-08-15
> **Answer to Reviewer 4Wwh**
>
> Thank you for your positive feedback on our work. We are glad to hear that you recognize the significance of our work and the potential usefulness of our multimodal dataset for estimating lower-body pose estimation. We hope that our dataset will indeed be helpful to researchers in this field. We appreciate your time and effort in reviewing our paper and look forward to contributing further to this area of research.

---

### Official Review · Reviewer_z3mb · 2023-07-27
**Valuable multimodal dataset for 3D body motion estimation**

**Rating:** 8
**Confidence:** 3

**Strengths:**

1. The collected multimodal dataset could be applied to healthcare applications.
2. The baseline model shows good generalization for 3D pose estimation on unseen exercises.

**Additional Feedback:**

Please refer to the above comments.

**Clarity:**

The paper is somewhat clear, but some important details need some effort to understand.

**Correctness:**

The collection of the dataset involves the capture of data from multiple sensors, which may encounter various issues, such as keypoint displacement for wearable devices or occlusion for cameras. By utilizing information from different modalities, methods can be designed to cross-validate the reliability and accuracy of the data.

**Documentation:**

The supplementary materials provides sufficient details for the proposed dataset.

**Ethics:**

No, I do not suspect it.

**Limitations:**

Since the dataset not only collects body movement but also muscle pressure, it should have a wider range of applications beyond 3D pose estimation. However, there was no relevant experiment or demonstration in the article.
The authors compared the results on 3d pose estimation with ground truth and get an average error of 7.21 degrees. However, it is not clear how good are these results as no comparable results were presented. Since this paper claims the proposed dataset can overcome some challenges faced by vision-based methods, some comparisons of qualitative or quantitative experiments on joint prediction should be provided.

**Opportunities For Improvement:**

1. The types of exercise could be more diverse and varied.
2. As the IMU data and the pressure data are two completely different modalities, it is sub-optimal to simply concatenate these inputs directly for fusion.

**Relation To Prior Work:**

The paper contributes some new ideas and represents incremental advances.

**Summary And Contributions:**

This paper collects a multimodal dataset and builds a baseline model to predict 3d joints on the lower body, in order to avoid some existing challenges from vision-based 3d pose estimation. This dataset involves pressure data, IMU data, and Mocap data captured by wearable knee sleeves, IMUs, and a Mocap system, respectively. The experiments show good performance on 3d pose estimation and generalization to unseen exercises.

---

> ### Author Response · Authors · 2023-08-15
> **Answer to Reviewer z3mb**
>
> Dear reviewer, we express our sincere gratitude for your insightful comments, which are invaluable in enhancing our manuscript. Please find our responses to your comments below:
> - Thank you for your insights regarding our activity types. We are indeed expanding our data collection to encompass a greater diversity of positions, including transitions from seated to standing and lying to seated. Presently, we're gathering data in gym settings, capturing activities such as running on a treadmill and squatting with weights. Moving forward, our aim is to continuously enhance our dataset's diversity to further validate our equipment's capability in accurately capturing poses across diverse scenarios.
> - We've opted to concatenate IMU sensor data and pressure sensor data due to their aligned trends. Our 9-axis IMUs generate data from accelerometers, gyroscopes, and magnetometers (abbreviated as acc, gyro, and mag in subsequent context). Notably, what we concatenate with the pressure sensor data are the quaternions, which are the result of fusing acc, gyro, and mag data. Prior to training, we normalize the pressure sensor data to a range of -1 to 1, matching the range of the quaternions. Additionally, the trends exhibited by the quaternions mirror those of the pressure sensor data, as muscle pressure correlates with human motion. Given these observations, our baseline approach integrates quaternions from the IMU with data from the pressure sensors directly for training. Yet, hierarchical networks that treat IMU and pressure data independently might also exhibit strong performance. In our multimodal dataset, we offer raw data from the acc, gyro, and mag sensors to support the needs of training diverse neural network architectures in the future.
> - Our multimodal dataset supports a wide range of applications, from home fitness and sports to healthcare and physical rehabilitation. It can also be utilized in ergonomics, prosthetics design, gaming, human-robot interaction, biomechanics research, wearable tech development, and more. We are extending these applications to include digital health and wellness experiences for musculoskeletal patients, both at home and in clinical settings. We add this statement to our revised manuscript (P2, lines 33-36, and P5, lines 134-139). Our wearable device is designed to promote healthcare applications, assist individuals with joint-related illnesses, and aid in sports analysis.
>
> - We initially refrained from presenting comparisons as our work is pioneering in providing a pressure sensor-embedded smart Knee Sleeve for joint rotation estimation and human pose tracking. Existing knee sleeve products and related datasets primarily focus on data from IMUs around knee joints, limiting their scope to knee joint prediction/calculation. In contrast, our work enables lower-body human pose estimation with eight joints. Other similar works either rely on non-integrated IMUs for pose estimation or estimate partial human poses using different types of equipment.
>
> - Our goal is to develop stretchable, user-friendly devices suitable for long-term outdoor use. We appreciate your thoughtful suggestion regarding the need for more comparisons to highlight our contributions. We have included Table A1 in our collective response to all reviewers, which compares our device with a range of wearable devices, highlighting our innovations. This table includes comparisons with works using only IMUs, textile, and sensor systems fused with IMUs and other flexible sensors. We emphasize our device's simplified setup, cost-effectiveness, and robustness, and its potential for more challenging tasks such as dancing and home fitness.
>
> - We are currently developing corresponding sleeves for full body suits and other wearable devices to capture comprehensive data on muscle and joint movements during human motion. We invite you to stay updated with our progress and look forward to sharing more of our work on a wider range of tasks in the future. Your valuable feedback is greatly appreciated.
>
> We genuinely appreciate your feedback. We hope our responses have adequately addressed your concerns. Should you have any further comments or suggestions, they would be most welcome and highly valued.

---

### Official Review · Reviewer_kVDB · 2023-07-28
**Lower body motion tracking using sensory sleeves and IMUs**

**Rating:** 6
**Confidence:** 5
**Correctness:** Yes, the claims are correct.
**Clarity:** The paper is well written but needs m…

**Strengths:**

This motion tracking dataset includes two sensing modalities and provides a new view of human activity monitoring.

**Additional Feedback:**

The website of the dataset can be improved. Adding some data visualization and videos could be helpful.

**Documentation:**

Yes

**Ethics:**

I don't see major ethical concerns.

**Limitations:**

I didn't see potential negative societal impact.

**Opportunities For Improvement:**

In terms of motion tracking performance, it is not clear regarding the advantages of using both IMUs and pressure sensors. The initial calibration is also difficult and it seems that the performance heavily relies on the initial pose. There is also no comparison with other motion tracking solutions using wearable sensors and the paper only describes the advantages over camera-based systems. More previous works should be mentioned in Section 2. There are many well developed IMU-based human full body motion tracking approaches. Some need dense IMUs, such as Xsens 3D (https://www.movella.com), and some need sparse IMUs (6 IMUs), like Sparse inertial poser by Marcard et al. 2017, Deep inertial poser by Huang et al. 2018, and Physical initial poser by Yi et al. 2022. In terms of the human body motion tracking, these approaches use simpler setup and can track full body motion with only two more IMUs than the setup proposed in this paper. Intuitively, more modalities are possible to provide more robust performance and even do more tasks. I recommend doing more detailed comparison for human body tracking and exploring other potential applications. And it should be possible to attach more devices to the upper body to do full-body motion tracking.

**Relation To Prior Work:**

The paper needs to mention and compare with more previous works of human body tracking using wearable sensors in addition to camera-based approaches.

**Summary And Contributions:**

This work includes a set of wearable hardware (sleeves for pressure sensing and IMUs), a dataset with ground truth labels from a MoCap camera system, and a machine learning pipeline to estimate lower body human pose. Different from previous works, the new motion capture approach relies on two sensing modalities, IMU motion data and muscle pressure measured by sleeves. A wearable device is attached to each leg to provide 14 channels of pressure data and 9 channels of IMU data. The experiment results from the proposed baseline model shows good performance.

---

> ### Author Response · Authors · 2023-08-15
> **Answer to Reviewer kVDB**
>
> We appreciate your feedback and have made revisions based on your insights.
> - Our study harnesses IMUs and pressure sensors for enhanced data accuracy in pose estimation tasks. IMUs, despite their efficacy, can experience drift due to small errors that accumulate. On the other hand, pressure sensors are adept at detecting minor joint movements, capturing deformations in skin and muscle tissues. Their data, when combined with our algorithm, presents a thorough understanding of the poses. The pressure sensors supplement IMUs, especially around the knee, by detecting muscle activity in the thigh and shank, aiding predictions of the hip and ankle joints. Furthermore, both IMUs and pressure sensors offer reliable kinesthetic data under various conditions. For a comparison of the combined IMU and pressure sensor data versus individual data, please see Table A2.
> - We agree that our current model is somewhat influenced by the initial positions when assessing the model's ability to generate unseen tasks. This issue can be mitigated through more standardized initial calibration. We propose two initial calibration methods in our paper: 1). Enforcing the initial position to be a standing pose. Even when data collection is intended from a seated pose, we request participants to initially stand and then transition to a sitting pose. 2). Modifying the IMUs to operate in absolute mode during the hardware setup (lines 263-268, P9 of the main context). Currently, the IMUs operate in relative coordinate mode, recording zero for initial data and calculating the rotation change during exercise. However, given that we are using 9-axial IMUs, we can detect the absolute angles at the starting point instead of using relative angles. The influence of initial poses on model performance is due to the inconsistency between the measuring modes of IMUs, pressure sensors, and Mocap. For instance, in seated poses, both the pressure sensor and Mocap's measurements operate in absolute mode, recording 90 degrees on knee angles as the initial state. In contrast, IMUs operate in relative mode, recording 0 degrees as the initial state. This inconsistency introduces errors in model prediction. The influence becomes more noticeable if the task is unseen by the model. To address this, we plan to implement absolute mode for all future data collection to maintain consistency between all sensors and IMUs. Simultaneously, we aim to collect more data with seated, standing, and lying positions as initial poses to create a more diverse dataset. In summary, while the issues of initial calibration and initial pose may seem challenging, they can be largely alleviated by ensuring consistent operation across all sensors and IMUs.
> - Thanks for sharing these insightful papers. All of them have exceptional contributions to our understanding of the field. We have incorporated their findings into our manuscript in the related work section and comparison table of our paper (P5 lines 117-139 and Table A1). We are exploring additional applications that could potentially benefit from our device. However, it's important to note that the aforementioned papers primarily focus on the use of individual IMUs attached to separate parts of the body for pose estimation and human movement analysis. In contrast, our work emphasizes the use of integrated wearable and stretchable devices embedded with pressure sensors. This approach allows us to gather more accurate data with fewer constraints, which enhances the precision of our analyses (Please see Table A2 for the reply to all reviewers on how pressure sensors contribute to the model accuracy). Once our model is trained, there is no longer a need for the supervision of the Mocap system, resulting in a simpler and more user-friendly setup. As for the initial calibration protocol we previously discussed, it can be streamlined by focusing on the collection of more diverse data by the developers. From a user experience perspective, our aim is to develop client-friendly applications that eliminate the need for attaching multiple sensors to various body parts, a process that could potentially hinder daily activities. Our focus is on creating stretchable, comfortable devices for frequent wear. We value your insightful suggestions about the need for comparative analysis to highlight our contributions. We have included a table comparing similar wearable devices to highlight our innovation in Table A1. Due to the current length limit of our manuscript, we will further explore more approaches in the camera-ready version.
> - Thank you for your suggestions regarding full-body estimation. We are developing corresponding full-body suits and other wearable devices to capture comprehensive data on muscle and joint movements during human motion. Please stay tuned for our future publications covering a wider range of tasks!
> - We will add videos of the 3D human model and multimodal data visualization to our website to make it more readable.

---

### Author Response · Authors · 2023-08-15
**Collective response to all reviewers**

We thank all reviewers for their insightful comments, which have been instrumental in enhancing the quality of our manuscript. We have observed that there are two recurring themes in the feedback provided, which we are keen to address. For a more comprehensive understanding of the tables and to address the unique concerns of each reviewer, we kindly request you to refer to our individual responses to each reviewer's comments. When referencing citations and serial numbers later in this response, please refer to the ones provided here.

According to all reviewers' feedback:

- We attach Table A1 on page 4 of the updated Appendix, with its associated explanations on lines 37-41 of page 3. Table A1 offers a comprehensive overview of our work in relation to current advancements, helping to contextualize our contributions in the broader field.
- We attached Table A2 with explains on lines 41-56 on page 4 of the Appendix, to elucidate the contribution of each modality within our multimodal dataset, emphasizing its impact on the overall accuracy of our system.
- We have expanded Section 2 to include a more comprehensive review of the established IMU-based motion tracking approaches and potential applications beyond 3D pose estimation (lines 117-139, P5 of the main manuscript). We believe that this addition not only provides a richer context to our work but also emphasizes the unique contributions of our study in the light of existing methodologies.
- We add more videos and details about our project on the public repository of [Github](https://github.com/Zhang-Wenwen/IntelligentKneeSleeves).

We include both the revised version and a version with updates highlighted in blue within the supplementary materials ZIP file.

- **Table A1: Comparative Analysis of Our Study vs. Related Research.**

|  | Sensors  Type | Integrated  Device | Wireless  Steaming | Task | Multi-person  scenario issue | Avg  RMSE |
|:---:|:---:|:---:|:---:|:---:|:---:|:---:|
| Our work | IMUs,  textile | y | y | Joint orientation  inference | n  | 7.21 deg (Avg angle  error) |
| Luo et al. (2021) [1] | Textile | y | n | Pose  classification | n | na* |
| DelPreto et al. (2022) [2] | IMU,EMG  tactile, camera | n | n | Activity  classification  in ketch | n | na |
| Luo et al. (2021) [3] | Tactile | y | n | 21 keypoint  estimation,  activity classification | y | 6.9 cm  (Avg location  error) |
| Tan et al. (2022) [4] | IMUs | n | nm* | knee flexion/extension  prediction | n | 9.52 deg  (Avg angle  error) |
| Huang et al. (2018) [5] | IMUs | n | nm | Pose estimation | n | 15.84 deg  (Avg angle  error) |

*nm: not mentioned.     *na: not applicable.

---------------------------------------------------------------------------------------------------------------------------------------------------------------------------------------

- **Table A2: The contributions of each modal to the prediction task.**

| Training Data | LHip | LKnee | LAnkel | LToe | RHip | RKnee | RAnkel | RToe |
|:---:|:---:|:---:|:---:|:---:|:---:|:---:|:---:|:---:|
| IMUs + Pressure sensor | 9.03 | 11.8 | 6.23 | 3.81 | 9.31 | 7.69 | 7.04 | 2.77 |
| Pressure sensor | 14.06 | 15.76 | 15.60 | 4.80 | 13.32 | 14.54 | 8.01 | 5.49 |
| IMUs  | 11.76 | 11.17 | 14.80 | 4.79 | 10.85 | 11.29 | 6.63 | 5.54 |

---------------------------------------------------------------------------------------------------------------------------------------------------------------------------------------

1.	Luo, Y. et al. Learning human–environment interactions using conformal tactile textiles. Nature Electronics 4, 193–201 (2021).
2.	DelPreto, J. et al. ActionSense: A multimodal dataset and recording framework for human activities using wearable sensors in a kitchen environment. in Advances in Neural Information Processing Systems (eds. Koyejo, S. et al.) vol. 35 13800–13813 (Curran Associates, Inc., 2022).
3.	Luo, Y. et al. Intelligent carpet: Inferring 3D human pose from tactile signals. in 2021 IEEE/CVF Conference on Computer Vision and Pattern Recognition (CVPR) 11255–11265 (IEEE, 2021).
4.	Tan, J.-S. et al. Predicting Knee Joint Kinematics from Wearable Sensor Data in People with Knee Osteoarthritis and Clinical Considerations for Future Machine Learning Models. Sensors  22, (2022).
5.	Vargas-Valencia, L. S. et al. Sleeve for Knee Angle Monitoring: An IMU-POF Sensor Fusion System. IEEE J Biomed Health Inform 25, 465–474 (2021).
6.	Jiang, J. et al. AvatarPoser: Articulated Full-Body Pose Tracking from Sparse Motion Sensing. in Computer Vision – ECCV 2022 443–460 (Springer Nature Switzerland, 2022).
7.	Hernandez, V., Dadkhah, D., Babakeshizadeh, V. & Kulić, D. Lower body kinematics estimation from wearable sensors for walking and running: A deep learning approach. Gait Posture 83, 185–193 (2021).
8.	Huang, Y. et al. Deep inertial poser: learning to reconstruct human pose from sparse inertial measurements in real time. ACM Trans. Graph. 37, 1–15 (2018).

---

### Decision · Program_Chairs · 2023-09-22

**Decision:**

Accept (Poster)

**Comment:**

This submission introduces a multi-modal dataset, collected with Intelligent Knee Sleeve. 3D GT of humans are collected with a synchronized motion capture system. The authors presented a baseline for estimating 3D human pose solely from wearable devices, supervised with 3D GT from the motion capture system. The 3D human pose estimation system does not suffer from visually challenging cases, such as occlusions. Potential applications of the proposed dataset are in rehabilitation. Overall, the authors provided extensive analysis of the dataset and baselines. All reviewers are positive about the submission.